# Early Signs of Molecular Defects in iPSC-Derived Neural Stems Cells from Patients with Familial Parkinson’s Disease

**DOI:** 10.3390/biom12070876

**Published:** 2022-06-23

**Authors:** Elissavet Akrioti, Timokratis Karamitros, Panagiotis Gkaravelas, Georgia Kouroupi, Rebecca Matsas, Era Taoufik

**Affiliations:** 1Laboratory of Cellular and Molecular Neurobiology—Stem Cells, Hellenic Pasteur Institute, 11521 Athens, Greece; akrioti@pasteur.gr (E.A.); panosgaravelas@hotmail.com (P.G.); gkouroupi@pasteur.gr (G.K.); rmatsa@pasteur.gr (R.M.); 2Bioinformatic and Applied Genomics Unit, Hellenic Pasteur Institute, 11521 Athens, Greece; tkaram@pasteur.gr

**Keywords:** Parkinson’s disease, alpha-synuclein, induced pluripotent stem cells, neuronal precursor cells, synaptogenesis

## Abstract

Parkinson’s disease (PD) is the second most common neurodegenerative disorder, classically associated with extensive loss of dopaminergic neurons of the substantia nigra pars compacta. The hallmark of the disease is the accumulation of pathogenic conformations of the presynaptic protein, α-synuclein (αSyn), and the formation of intraneuronal protein aggregate inclusions. Neurodegeneration of dopamine neurons leads to a prominent dopaminergic deficiency in the basal ganglia, responsible for motor disturbances. However, it is now recognized that the disease involves more widespread neuronal dysfunction, leading to early and late non-motor symptoms. The development of in vitro systems based on the differentiation of human-induced pluripotent stem cells provides us the unique opportunity to monitor alterations at the cellular and molecular level throughout the differentiation procedure and identify perturbations that occur early, even at the neuronal precursor stage. Here we aim to identify whether p.A53T-αSyn induced disturbances at the molecular level are already present in neural precursors. Towards this, we present data from transcriptomics analysis of control and p.A53T-αSyn NPCs showing altered expression in transcripts involved in axon guidance, adhesion, synaptogenesis, ion transport, and metabolism. The comparative analysis with the transcriptomics profile of p.A53T-αSyn neurons shows both distinct and overlapping pathways leading to neurodegeneration while meta-analysis with transcriptomics data from both neurodegenerative and neurodevelopmental disorders reveals that p.A53T-pathology has a significant overlap with the latter category. This is the first study showing that molecular dysregulation initiates early at the p.A53T-αSyn NPC level, suggesting that synucleinopathies may have a neurodevelopmental component.

## 1. Introduction

Parkinson’s disease (PD) and related synucleinopathies are a group of incurable neurodegenerative disorders associated with alpha-synuclein (αSyn) pathology [1]. αSyn is the major sporadic PD-linked gene [2] while point mutations and multiplications of the locus [3,4,5] cause autosomal dominant forms of early onset and aggressive Parkinsonism and dementia. Despite intensive research efforts using animal and cellular models, the mechanisms of disease, both sporadic and familial, remain unresolved. Recent technological advancements with induced pluripotent stem cell (iPSC)-based models derived from somatic cells of patients with sporadic or familial PD [6,7], offer great opportunities to elucidate disease phenotypes and investigate the underlying mechanisms and screen for new drugs in a human setting. Morphometric and functional analyses of live-patient neurons from familial PD cases [8,9,10,11] have shown the presence of numerous disease-associated phenotypes such as the presence of intraneuronal protein aggregates, compromised neuritic growth, αSyn- and Tau-associated axonal pathology, mitochondrial and ER stress, and reduced synaptic connectivity. Collectively these neuropathological features closely resemble hallmarks previously identified in post-mortem brains of the respective disorders [12] demonstrating that human iPSC-derived models can recapitulate human pathology and serve as valuable models for both basic and translational research. Most important is that iPSC-based studies of familial cases of PD and other degenerative pathologies indicate that the molecular and cellular alterations that eventually lead to extensive neuronal loss seem to be initiated much earlier than thought, indicating a potential neurodevelopmental component in PD. The first clue came from the identification that human neurons in the dish that exhibit degenerative signs resemble human fetal neurons based on their immature molecular identity, as revealed from sequencing approaches [13]. Second, synaptic dysfunction is identified as an early prominent feature in these systems that exhibits excitatory–inhibitory imbalance and poor formation of neuronal networks, both highly similar to those identified in models of neurodevelopmental disorders [8]. Third, compounds that are currently used for improving cognitive defects in AD and PD were highly effective in restoring neurotransmission in autism-derived human neurons [14,15]. Finally, the limited data we have on brain organoids from PD and AD cases clearly demonstrate the aberrant formation of early neuronal networks accompanied by the appearance of disease-associated phenotypes [16,17,18]. The traditional dichotomization of early-onset neurodevelopmental and late-onset neurodegenerative diseases is currently challenged. Here we propose that aberrant molecular and cellular processes are already present in early neuronal precursors (NPCs) derived from iPSC of familial PD patients that carry the G209A mutation in the αSyn gene SNCA, resulting in the pathological p.A53T-αSyn protein expression [19]. At this stage, mutant αSyn transcript expression leads to gene expression changes related to cell differentiation and metabolic processes, alters the cell adhesion pathways required for synaptogenesis and axonal guidance, affects early synaptic markers, and impairs the synaptogenic ability of neurons. This report shows that pathological αSyn expression has negative effects at the early stages of neuronal differentiation contributing to the establishment of dysfunctional neuronal networks and gradual neurodegeneration. Whether these molecular defects are causative or contributors to PD pathology is not known. However, their characterization is essential for understanding PD pathophysiology and increasing the translatability of pre-clinical studies in PD and other related NDs.

## 2. Materials and Methods

### 2.1. Generation of Human iPSC-Derived NPCs

iPSCs lines from a p.A53T patient (PD) and an age and sex-matched control subject (CTR) were derived, under the ethical permission of the Scientific Council and Ethics Committee of Attikon University Hospital (Athens, Greece) and the Hellenic Pasteur Institute Ethics Committee, as previously described [8] (Appendix A). Briefly, skin fibroblasts from a 47-year-old male patient and an age and sex-matched control subject were acquired from the Biobank generated within the European Project on Mendelian Forms of Parkinson’s Disease (MEFOPA). iPSCs were generated by transduction of human fibroblasts with retroviral vectors expressing the human cDNAs of OCT4, SOX2, KLF4, and C-MYC, and all lines used for differentiation demonstrated normal karyotypes. For the generation of NPCs, iPSCs were dissociated with accutase (Life Technologies, Framingham, MA, USA) for 10 min and re-suspended in embryoid body (EB) medium consisting of KnockOut DMEM (KO-DMEM, Life Technologies), 20% Knockout Serum Replacement (KSR, Life Technologies), 2 mM GlutaMax, MEM Non-Essential Amino Acids (100× MEM NEAA, Life Technologies), and 100 mM β-mercaptoethanol (Life Technologies), supplemented with 10 µM ROCK inhibitor Y-27632 (Tocris Bioscience, Bristol, UK) on non-adherent Petri dishes. After 5 days in vitro (DIV), neural induction was achieved by dual suppression of the SMAD signaling pathway using a combination of Noggin (250 ng/mL; R&D Systems, Mineapolis, MN, USA) and SB431542 (10 µM; Tocris Bioscience) in DMEM:F12/N2-medium (neural induction medium) onto poly-L-ornithine (PLO; 20 µg/mL; Sigma-Aldrich, Burlington, MA, USA)/laminin (5 µg/mL; Sigma-Aldrich)-coated coverslips for 8 DIV to generate NPCs. NPCs were either expanded in STEMdiff™ Neural Progenitor Medium (STEMCELL Technologies, Vancouver, BC, Canada) or were differentiated spontaneously in DMEM:F12/B27/N2-medium for 10–15 DIV.

### 2.2. RNA Sequencing and Analysis

Total RNA from CTR and PD NPCs was extracted and sequenced on an Illumina HiSeq sequencer at the European Molecular Biology Laboratory (EMBL) as previously described [8]. Analysis was performed by the Bioinformatics Unit of HPI. A detailed protocol is included in Appendix A. RNA-Seq data have been deposited in Gene Expression Omnibus (accession no. GSE84684).

### 2.3. RNA Isolation, cDNA Synthesis, and qPCR

Total RNA was extracted from cell pellets using the TRIzol^®^ Reagent (Life Technologies). Following digestion with DNase I, 1 µg of total RNA was used for first strand cDNA synthesis with the ImProm-II Reverse Transcription System (Promega, Madison, WI, USA) according to the manufacturer’s instructions. Quantitative RT-PCR was performed in a LightCycler 96 Instrument (Roche, Basel, Switzerland) and the analysis of relative gene expression was based on the comparative CT Method (2^−ΔΔCT^ Method) (Appendix A). All primers used are listed in Appendix A.

### 2.4. Immunofluorescence Staining

Cells were fixed with 4% paraformaldehyde (Sigma-Aldrich) for 20 min at room temperature. Samples were blocked with 0.1% Triton X-100 (Sigma-Aldrich) and 5% donkey serum in PBS for 30 min and were subsequently incubated with the primary antibodies: anti-NESTIN (1/200; Merck-Millipore, Burlington, MA, USA ABD69), anti-PAX6 (1/100; DSHB, Douglas, Houston, TX, USA AB 528427), anti-α-Synuclein (αSYN; 1/500; BD Biosciences, Franklin lakes, NJ, USA 610787), anti-βIII-tubulin (TUJ1, 1/1000; Cell Signaling, Danvers, MA, USA 5568), anti-Synapsin 1 (1:200; Cell Signaling), and anti-HA-probe (1:500; Santa Cruz, Dallas, TX, USA) at 4 °C overnight, followed by incubation with appropriate secondary antibodies (Molecular Probes, Thermo Fisher Scientific, Waltham, MA, USA) conjugated to AlexaFluor 488 (green) or 546 (red), for at least 1 h at room temperature. Coverslips were mounted with ProLong Gold antifade reagent with DAPI (Cell Signaling) and images were acquired using a Leica TCS-SP8 confocal microscope (LEICA Microsystems) and analyzed using ImageJ software (version 1.53c) (NIH).

### 2.5. Artificial Synapse Formation Assays and Synapse Induction Analysis

Artificial synapse formation assays were performed as previously described [20] with modifications. A detailed protocol is included in Appendix A.

## 3. Results

### 3.1. Generation and Characterization of p.A53T-αSyn NPCs

PD patient-derived NPCs carrying the SNCA G209A mutation (PD) and those from an unaffected control (CTR) were derived from iPSCs as previously described [8], following a neural induction protocol based on dual-SMAD inhibition. At this stage, CTR and PD NPCs expressed similar numbers of both Nestin (intermediate filament protein; neural progenitor cell marker) and Pax6 (transcription factor; marker for early neuronal differentiation) (Nestin+: CTR 79.24 ± 6.23% and PD 84.21 ± 5.75%; PAX6+: CTR 72.01 ± 9.04% and PD 75.36 ± 7.88%, n = 3) (Figure 1A) and this was further confirmed by RT-qPCR (Figure 1B). At this differentiation time point, only a small fraction of cells expressed βΙΙΙ-tubulin+ (TUJ1+: CTR 8.6 ± 2.48% and PD 10.12 ± 1.89%) and MAP2+ (>4% in both CTR and PD cultures), while GFAP+ cells (glial fibrillary acidic protein; astrocytic marker) were not detected. Subsequently, cells were differentiated spontaneously into βΙΙΙ-tubulin+ (TUJ1+) neurons for two weeks to assess levels of αSyn in early neurons (Figure 1C,D). Immunofluorescence revealed that α-Syn was present in the soma and neurites of both PD and control TUJ1+ neurons, though more cells were strongly positive for α-Syn in PD cultures (Figure 1C). In agreement, quantification of αSyn mRNA by RT-qPCR revealed elevated levels of this transcript at this differentiation stage (Figure 1D). As previously reported in both mouse and human neurons, αSyn showed a selective and fluctuating expression in TUJ1+ cells. However, the functional significance of this phenomenon is still not defined [11,21].

### 3.2. Comparative Gene Expression Profiling of p.A53T-αSyn NPCs

To identify genes whose expression is affected by the expression of mutant αSyn at the NPC stage, transcriptome-wide RNA-seq analysis was performed at a specific stage of the directed differentiation procedure corresponding to iPSC-derived NPCs (13 DIV) [8] (Figure 1A,B). Total RNA from two control lines (CTR1 and CTR2) and two PD lines (PD1 and PD2) was utilized for cDNA library preparation. Following poly-A selection, RNA-seq was performed for global gene expression profiling. Analysis of protein-coding genes revealed 324 differentially expressed genes (DEGs) between sex and age-matched PD and CTR samples. In particular, 100 transcripts were downregulated and 224 upregulated (*p* value ≤ 0.05 and log (fold change)) (Appendix A). To unravel molecular mechanisms and signaling networks associated with αSyn expression in NPCs, we categorized DEGs according to molecular function, biological processes, and cellular components based on ontology analyses (Appendix A).

To obtain a more comprehensive understanding of how the differential expression of DEGs is linked to the expression of the mutant αSyn in iPSC-derived NPCs, we combined, by manual meta-analysis, the GO term enrichment analysis with DAVID functional annotation clustering, and information derived from Hyperlinked Over Proteins (iHOP) and PubMed–National Center for Biotechnology Information. Based on our previous work showing extensive synaptic dysfunction of PD neurons [8], we focused on DEGs within main annotation clusters related to metabolism, cell differentiation and development, cytoskeletal organization, ion channels and transporters, cell adhesion and extracellular matrix (ECM), and synapse organization and have validated selected transcripts from each category by qPCR (mean expression mRNA levels of selected transcripts from RNA-seq analysis are listed in Appendix A).

*Metabolism.* The most prominent category affected by mutant αSyn expression in NPCs involves metabolic processes (50 genes: 23 downregulated and 27 upregulated) (Figure 2A, Appendix A). Within this cluster, lipid and amino acid biosynthesis and regulation transcripts, along with numerous oxidoreductases encoding genes were identified. Of particular interest were lipid biosynthesis components (ALOX15B, GLIPR1, CERS3, PON1, AGPAT9, SGSM2, RARRES2, LPAR6, ATP8A1, PLCG2, SULT4A1, PTGS1) that have been shown to influence energy sources and signaling entities [22]. Specifically, lipid-oxidizing enzymes such as cell proliferation and neurogenesis act as building blocks of membranes, alternatALOX15B (logFC: −3.45424) which produce fatty acid metabolites and modulate oxidative stress [22] lead to decreased neuronal concentrations of anti-oxidant glutathione levels, an early biomarker of PD. GLIPR1 (logFC: −2.5334) a master regulator of lipid biosynthesis, has been identified in PD blood samples as having a differential methylation state [23]. PLCG2 (logFC: 1.407053), a component of phospholipase C, has missense variants that have been identified as risk factors for NDs including PD [24]. LPAR6 (logFC: 1.287036), an LPA receptor that plays an essential role in CNS development [25] is also dysregulated in mutant NPCs. Interestingly, three members of the metallothionein family, MT1X (logFC: −1.55422) MT2A (logFC: −1.38683), and MT1F (logFC: −1.32454) that are involved in copper dyshomeostasis and alpha-synuclein aggregation [26] were significantly downregulated in PD NPCs. Validation of CA7 (logFC: 4.112885) mRNA levels (Figure 2A) confirmed that this intracellular carbonic anhydrase previously described to regulate neuronal pH buffering and actin dynamics [27] is highly upregulated in PD NPCs while CRYZ (logFC: −6.46325) and TYW3 (logFC: −8.48497), a quinone oxidoreductase, and an enzyme of wybutosine-tRNA(Phe) biosynthesis respectively [28], were almost absent in PD NPCs (Figure 2A).

*Cell Differentiation and Development.* The second-largest cluster affected by p.A53T-αSyn expression contains genes involved in cell differentiation and development. Even though the comparison of NPC markers did not reveal any neuroectodermal acquisition defects as previously described in HD and AD NPCs [16,29,30,31], the transcriptome profile of mutant NPCs indicates differences in 32 factors (26 upregulated and 6 downregulated) (Figure 2B, Appendix A) including the neurogenic molecules, NeuroD1, NeuroG1, NeuroG2, EDN1, CHRDL1, ERBB3, EFNA2, SOX8, LGI4, NTRK1, ISLR2, FOXD3, DCDC2, FOXN4, and NTN5. Specifically, the mRNA levels of NEUROD1 (logFC: 3.046852), a bHLH transcription factor that plays an important role during neuronal differentiation and is instrumental for iPSC conversion to NPCs [32,33], NEUROG1 (logFC: 3.012214) a neurogenic factor that induces widespread neuronal differentiation [34], and NEUROG2 (logFC: 1.409525), essential for neuronal commitment, cell cycle withdrawal, and neuronal differentiation [35], were found significantly affected in biological replicates (Figure 2B).

*Ion channels and transporters.* Neurophysiological processes depend largely on the correct spatiotemporal expression of channels and transporters that follows strict developmental programs [36]. NPCs express various types of voltage-gated K^+^, L^+^ type Ca^+^ channels, and Cl^−^ channels. The expression profile demonstrates that the expression of p.A53T-αSyn affects mainly the expression of potassium channels subunits (KCNE4, KCNJ15, KCNAB3, KCNQ2, KCNC1, KCNQ4, KCNQ1, KCNH6), including those of KCNC1 and KCNQ4 that are essential for neuronal responsiveness to synaptic inputs [37] (Figure 3A, Appendix A). In addition, Ca^+^ channels subunits and related signaling components (11) were also dysregulated in mutant NPCs, including CALB1 (logFC: −1.68846), involved in the regulation of post-synaptic cytosolic calcium ion concentration and SLC8A2 (logFC: 1.446658), a modulator of long-term synaptic potentiation (Figure 3A).

*Cytoskeleton.* Studying cytoskeletal dynamics during in vitro neurogenesis of iPSC has revealed both key players and respective protein modifications [38]. This finely tuned process defines later neuronal polarity, and mutations in essential mediators are associated with severe developmental brain abnormalities [39] while alterations in their acetylation are linked to AD, PD, and ALS [40]. Eighteen transcripts were altered by p.A53T-αSyn expression (5 upregulated and 13 downregulated) (Figure 3B, Appendix A), with the majority affecting actin dynamics (ACTG2, TAGLN, ACTA2, PDLIM3, NRK, SYNPO, THSD7B, SAMSN1). In addition, NRK (logFC: −1.58127), involved in neuronal projection morphogenesis [41], SYNPO (logFC: −1.08403), a modulator of actin-based shape and motility of dendritic spines, and JAKMPI1 (logFC: 2.379211), microtubule-associated transporter of GABA-B receptor [42] were significantly altered in PD NPCs (Figure 3B).

*Cell adhesion and extracellular matrix organization.* Studies focusing on how cell–ECM interactions affect neural stem cell developmental fate and specification have clearly demonstrated that even delicate changes in ECM components affect neural progenitor viability, differentiation, migration, and eventually neurite outgrowth of developing hiPSC-derived neurons. As with any other in vitro cell system, NPC’s culture is dependent on exogenous ECM substrates. However, as differentiation proceeds, NPCs produce a whole network of ECM proteins that not only helps their survival and cell-to-cell communication but is essential for neuronal differentiation, maturation, and synaptogenesis. At the NPC stage, 25 genes involved in cell adhesion were differentially expressed in PD NPCs from which 5 encode for proteins involved in the guidance of migrating neurons and axons during development and synaptic plasticity (TNC (logFC: −1.9285)), promotion of neurite outgrowth (FLRT1 (logFC: 1.50722), binding to alpha-neurexins (NXPH3 (logFC: 1.688257)), and establishment and maintenance of specific neuronal connections in the brain (PCDHA6 (logFC: 3.897926), CDH23 (logFC: 1.497305)) (Figure 3C, Appendix A).

*Synapse formation and organization.* Synaptogenesis is the formation of a fully functional synapse that follows a specific pattern of events: initial axon–dendrite contact, synapse formation, and synapse maturation [43]. Even though the NPC stage after iPSC-directed neuronal differentiation might not seem the appropriate developmental time point to study synaptogenesis, mRNA levels of pre-, post-, and trans-synaptic adhesion molecules are already apparent and could be predictive of synaptogenesis defects detected at the next developmental stages. In context, our previous data in PD neurons showed defects at various steps of synaptogenesis that could be initiated at the NPC stage as demonstrated by the aberrant levels of 23 transcripts (20 upregulated and 3 downregulated) associated with synaptic organization and regulation (Figure 3D, Appendix A). From these, four transcripts encode for neuron-enriched adhesion molecules (SLITRK4, FLRT2, ROBO1, CADM2), five for synaptic vesicle recycling (SYCP2L, SYT7, DOC2A, SYNGR3, SLC6A3) and six for glutamatergic neurotransmission (GRID1, GRIP2, PRPT1, PRKG2, GRIK4, GRIN3B). The decreased expression of GRID1 (logFC: −1.36147), a glutamate ionotropic receptor subunit gene, and concurrent increase of GABRQ (logFC: 1.897966), a gamma-aminobutyric acid type A receptor, and SYT7 (logFC: 1.51335) encoding for synaptotagmin 7, that was recently shown to ensure high-frequency transmission at central GABAergic synapses [44], were confirmed in biological replicate samples (Figure 3D) and could be indicative of the neurotransmission imbalance in PD neurons.

*Transcripts associated with neurological disorders.* Similar to p.A53T-αSyn neurons, mutant NPCs had DEGs previously been implicated with mental disorders, developmental monogenic diseases, and Tau- and αSyn-related pathologies (Appendix A). GO enrichment analysis revealed transcripts associated with schizophrenia (CALN1, NEUROG1, GFRA2, PDYN, CADM2, GABRQ, GRIN3B, COMT, CDH15, SYT7) and other mental disorders, including bipolar and ASD. The two most downregulated mRNAs in mutant NPCs, CRYZ and TYW3, are the products of two linked genes on chromosome 1, with SNPs associated with ALS development [28]. CRYZ encodes for zeta-crystallin, nicotinamide-adenine dinucleotide phosphate-dependent quinone reductase, and TYW3 for tRNA-wybutosine synthesis protein 3 homolog, two metabolic proteins associated with insulin resistance, a clinical feature with a high prevalence to PD [45]. Associated with metabolism pathways are DEGs related to metabolic syndromes like maturity-onset diabetes of the young 6 (NEUROD1) [46], Sjogren-Larsson Syndrome (NXPH3), lipodystrophy (CAV1) [47], and Rett Syndrome (GRID1) [48]. Finally, transcripts known to be involved in Tau- and synucleinopathies include IGLON5, CAV1, GFRA2, SYNGR3, and SLC6A3, with the latter being particularly interesting as its genetic variants lead to Parkinsonism-dystonia infantile (PKDYS) syndrome [49].

DEGs in NPCs vs. neurons. The cellular heterogeneity and variability following hiPSC-dopamine neuron differentiation have been recently demonstrated (Fernandes et al., 2020) by unbiased single-cell RNA-Seq, where the concurrent presence of various subtypes of mature neurons and NPCs, even at a late stage of directed neuronal differentiation, were identified. As we wanted to exclude that differential expression of selected transcripts at the NPC stage might be due to the presence of the small percentage of TUJ1+ cells detected, we assessed the levels of progenitor and mature neuronal markers at both developmental stages by integrating data from our previous transcriptomic analysis performed in mature neurons (day 40–50) [8] (Appendix A). First, we confirmed the differential composition of cultures at each differentiation point. Specifically, mRNAs of NPC markers (NFIA, LMX1A, OTX2) were elevated at NPCs and dropped in mature neurons and vice versa, neuronal markers (DCX, MAP2, TUBB3, RBFOX3, TH, SYT1, SNAP25) increased dramatically as neurons matured, suggesting an enrichment of NPCs within cultures at the time point selected for the RNA-seq analysis in this study. The expression of genes included in the categories described in Figure 2 and Figure 3 was examined at the next differentiation stage where neurons are the prominent cell type present (Appendix A) revealing that only a limited number (20 transcripts) remained differentially expressed at this later phase. Overall, this comparative analysis demonstrates specific αSyn-induced effects on the NPC transcriptome.

### 3.3. Artificial Synapse Formation in p.A53T-αSyn Neurons

Our previous work on p.A53T-αSyn neurons showed synaptic dysfunction and aberrant neuronal network organization [8] that could be linked to the molecular changes observed in NPCs. Based on the abnormal mRNA levels of axon guidance, ion channels, and synaptic and cell adhesion genes, we hypothesized that mutant neurons might show compromised initial synapse formation potential. To address this, we exploited the artificial synapse formation (ASF) assay to investigate possible synaptogenesis defects in p.A53T-αSyn neurons. This method traditionally employs ectopic expression of post-synaptic molecules in HEK293T cells that permits the formation of artificial synaptic contacts when in co-culture with neuronal cells. Specifically, CTR and PD patient-derived neurons were differentiated for 2 weeks, replated for 3 days, and then co-cultured with HEK293T cells overexpressing either the post-synaptic molecule NLGN2 (inhibitory synapse) or NLGN4 (excitatory synapse) tagged to the HA epitope, while a plasmid expressing eGFP served as a negative control. Immunofluorescence analysis revealed the formation of Synapsin1/HA-positive puncta (Figure 4A) due to the expression of either NLGN2 (Figure 4b,e) orNLGN4 (Figure 4h,k). To semi-automatically quantify the synaptic connections in the different experimental conditions, a pipeline based on the co-culture index (C.I.) measurement [50] was used. Briefly, this approach counts the Synapsin1 density that is related to the target HEK293T cells and excludes random Synapsin1 distribution. HEK293T cells with CI ≥ 1 are considered positive cells. The first observation was that a large number of HEK293 cells (approximately 80%) exhibited Synapsin+ puncta on their surface when they expressed NLGN2-HA, and this number was equal between CTR and PD neuronal cultures. However, when NLGN4-HA was expressed, a significantly lower number of HEK293 cells showed Synapsin+ density when co-cultured with PD neurons (Figure 4B) which could represent a compromised capacity of PD neurons to recognize and get attracted to excitatory post-synaptic signals. The second finding comes from an analysis of the C.I. levels of HEK293 positive cells in different experimental conditions (Figure 4C). In the case of HEK293-NLGN2_HA positive cells, a higher index was identified in cultures with PD neurons suggesting that once they come in contact with the non-neuronal cells the density of Synapsin1 is increased, suggestive of an increased tendency to form inhibitory connections compared to CTR neurons. On the other hand, when they come in contact with cells expressing an excitatory post-synaptic molecule such as NLNG4, they form similar connections to CTR neurons. Overall, these data shows an impairment of PD neurons to initialize and maintain balanced synaptic connections that could correlate with the observed differential levels of transcripts that orchestrate this process at the NPC differentiation stage.

## 4. Discussion

The onset of motor and cognitive symptoms in PD and most related synucleinopathies is late. Consequently, as with other NDs, PD has been traditionally classified, investigated, and treated as an age-related degenerative pathology that still has no cure, disease-modifying strategies, and reliable diagnostic and predictive biomarkers. This view has been slowly changing due to advanced clinical tools that monitor subtle changes in patients long before diagnosis [51], and the identification of neurodevelopmental disturbances in mouse models and related in vitro cellular systems [18]. It was not though until the use of pluripotent stem cells used to create disease-in-a-dish models of CNS pathologies that the existence of a neurodevelopmental component in NDs was clearly suggested. These models are characterized by an inherent neurodevelopmental nature as differentiated neurons are immature [52]. This feature renders them ideal to study neurodevelopmental cellular and gene expression perturbations. Within this context, we have been focusing on elucidating the pathogenesis of PD by monitoring the molecular and cellular changes along the differentiation axis of p.A53T-αSyn “iPSC→NPCs→Neurons” to identify early molecular and cellular processes that could be linked to earlier disease onset. Our previous [8] and current work demonstrate synaptic defects in mutant neurons that already exist, at the mRNA level, in neuronal precursors. Here we demonstrate differential expression of numerous axon guidance, synapse formation and organization, differentiation, and developmental genes in mutant NPCs without parallel detection of gross morphological changes. This is in agreement with the sole report on how pathological αSyn affects human NPCs, where higher levels of αSyn due to triplication of the locus affected neuronal differentiation and maturation [53] while NPCs from patient and control lines were morphologically indistinguishable. However, our study lacks detailed morphometric analysis of finer neuronal structures including neurites, therefore we cannot exclude that changes at the mRNA level do correlate with morphological alterations. Remarkably, our data closely resemble the RNA-seq profile of HD NPCs where faulty neuronal determination and cell polarization were also revealed [30]. Similar to ASD and other mental disorders [54,55], the transcriptome profile of mutant NPCs shows increased levels of multiple transcripts associated with synaptogenesis and neuronal network formation. Indeed, the profile of p.A53T-αSyn NPCs contains multiple transcripts associated with developmental genetic diseases and ASD that could explain the defective capacity of neurons to form balanced inhibitory and excitatory connections in ASF conditions. Overall, the observed early appearance of synaptic dysfunction could be linked to deficits in specific developmental pathways in familial PD, however mechanistic studies for the identification of specific molecular targets in both 2D and 3D cell systems are required to address this hypothesis. Despite the lack of evidence from αSyn-based 3D systems, promising data comes from the generation of LRRK2, HD, and AD organoids that exhibit both severe developmental defects along with disease-associated phenotypes [18,30,56,57].

Except for the link between pathological aSyn, neuronal differentiation, and synaptogenesis defects, the transcriptome analysis of p.A53T-αSyn NPCs reveals biological processes that have not been considered to be at the core of early PD pathology, including lipid metabolism and cytoskeletal organization. It is well known that aSyn interacts with multiple lipids through its N-terminal domain with variable affinities. However, the physiological importance of such interactions in NPCs and their outcome in proliferation and neurogenesis remains to be investigated. Human neurogenesis is largely influenced by cytoskeletal organization and remodeling [38,58] and we know that iPSC neuronal differentiation during the first 30 days requires reorganization of actin filaments and finely regulated localization of actin-related proteins [59]. Interestingly, the majority of mRNA transcripts identified in this study encode for actin family members rendering further investigation of the relationship between aSyn and cytoskeletal organization at early stages of pathology particularly intriguing. Collectively this study reveals a wide dysregulation of the p.A53T-αSyn NPC transcriptome that could serve as the basis to identify specific molecular targets and related pathways in the future.

Studying precursor populations for NDs-associated phenotypes and mechanisms is still in its early steps and carries significant criticism regarding its value for the patients. However, basic knowledge of when PD initiates, if there is a neurodevelopmental component, and the definition of the spatiotemporal events that eventually lead to the clinical symptoms in PD patients are of immense translational value for the development of biomarkers and PD therapeutics.

## Figures and Tables

**Figure 1 biomolecules-12-00876-f001:**
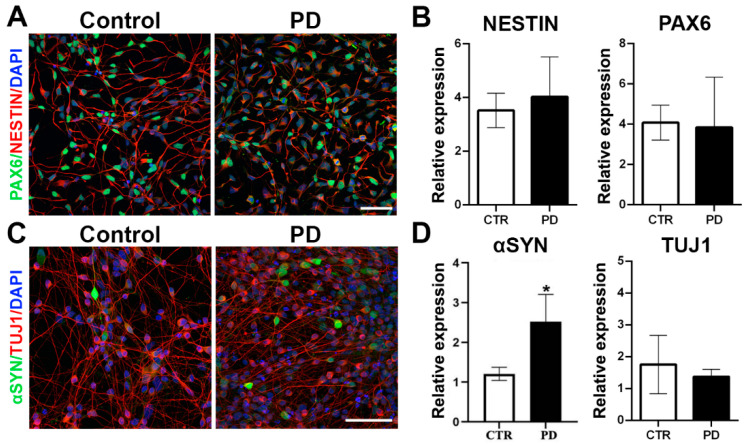
NPC characterization and spontaneous neuronal differentiation. (**A**). Immunostaining of CTR and PD iPSC-derived NPCs for PAX6 (green) and NESTIN (red). Cell nuclei are counterstained with DAPI (blue). Scale bar, 50 µm. (**B**). RT-qPCR analysis of NESTIN and PAX6 mRNA expression normalized to GAPDH levels. Data represent mean ± SEM (n = 3). (**C**). Immunostaining for α-synuclein (αSYN; green) and βIII-tubulin (TUJ1; red) in CTR and PD iPSC-derived neurons at 25 DIV. Cell nuclei are counterstained with DAPI (blue). Scale bar, 50 µm. (**D**). RT-qPCR analysis of mRNA expression for α-synuclein (αSYN) and βIII-tubulin (TUJ1). Data represent mean ± SEM (n = 3, * *p* < 0.05).

**Figure 2 biomolecules-12-00876-f002:**
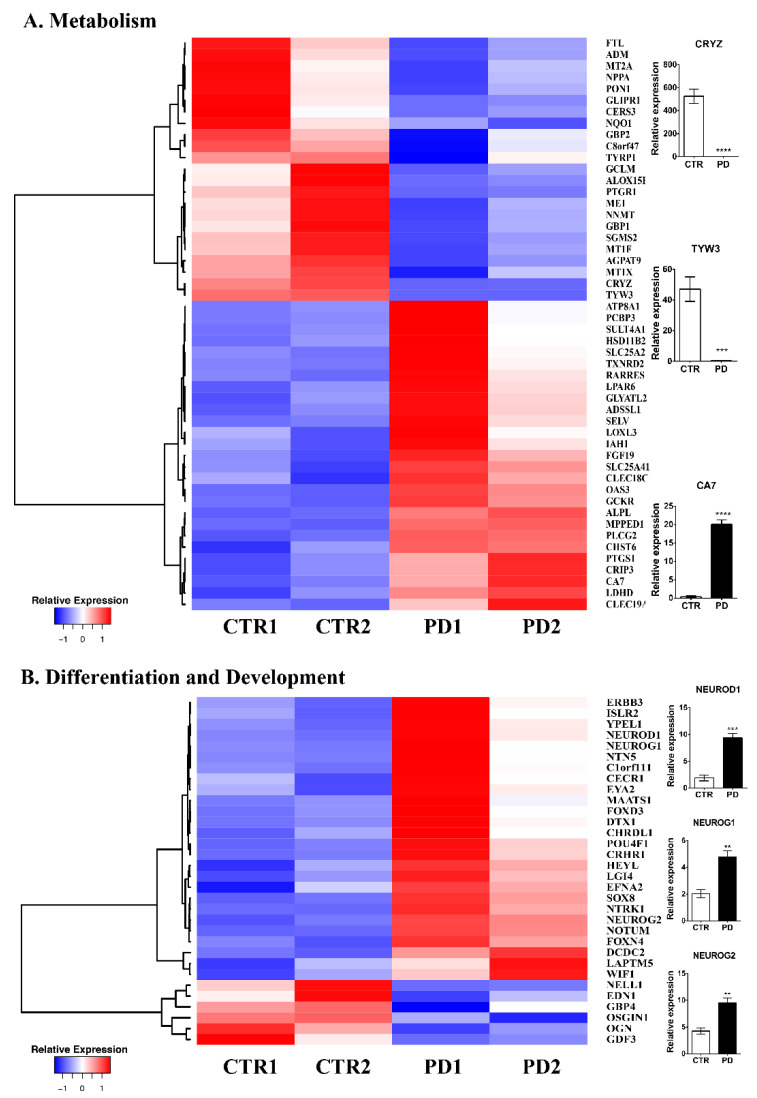
Gene expression analysis of iPSC-derived NPCs. (**A**,**B**) Differential gene expression between control (two clones from one healthy subject, CTR1 and CTR2) and PD (two clones from one patient, PD1 and PD2) iPSC-derived NPCs at 13 DIV. Heat maps of genes encoding metabolism (**A**) and cell differentiation and development genes (**B**). RT-qPCR analysis of selected genes (**A**,**B**) in control and PD NPCs. Gene expression normalized to GAPDH. Data represent mean ± SEM (one-way ANOVA, ** *p* < 0.01, *** *p* < 0.001, **** *p* < 0.0001, n = 3).

**Figure 3 biomolecules-12-00876-f003:**
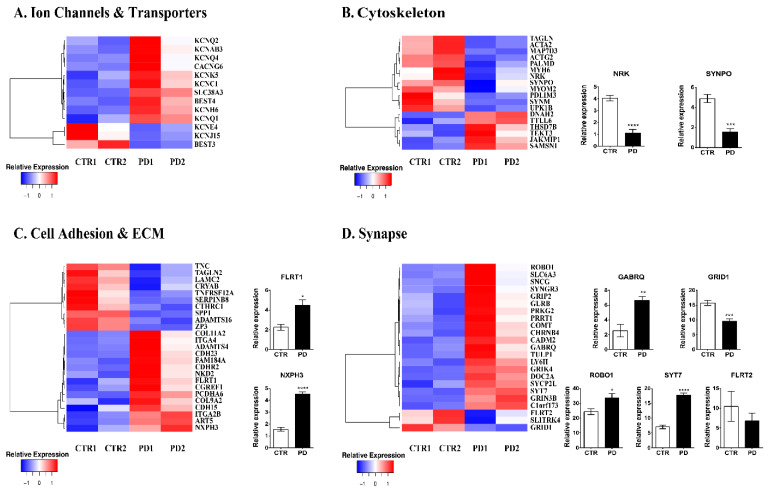
Gene expression analysis of iPSC-derived NPCs. (**A**–**D**) Differential gene expression between control (two clones from one healthy subject, CTR1 and CTR2) and PD (two clones from one patient, PD1 and PD2) iPSC-derived NPCs at 13 DIV. Heat maps of genes encoding ion channels and transporters (**A**), cytoskeleton (**B**), cell adhesion and ECM (**C**), synapse-associated genes (**D**). RT-qPCR analysis of selected genes 9 (**B**–**D**) in control and PD NPCs. Gene expression normalized to GAPDH. Data represent mean ± SEM (one-way ANOVA, * *p* < 0.05, ** *p* < 0.01, *** *p* < 0.001, **** *p* < 0.0001, n = 3).

**Figure 4 biomolecules-12-00876-f004:**
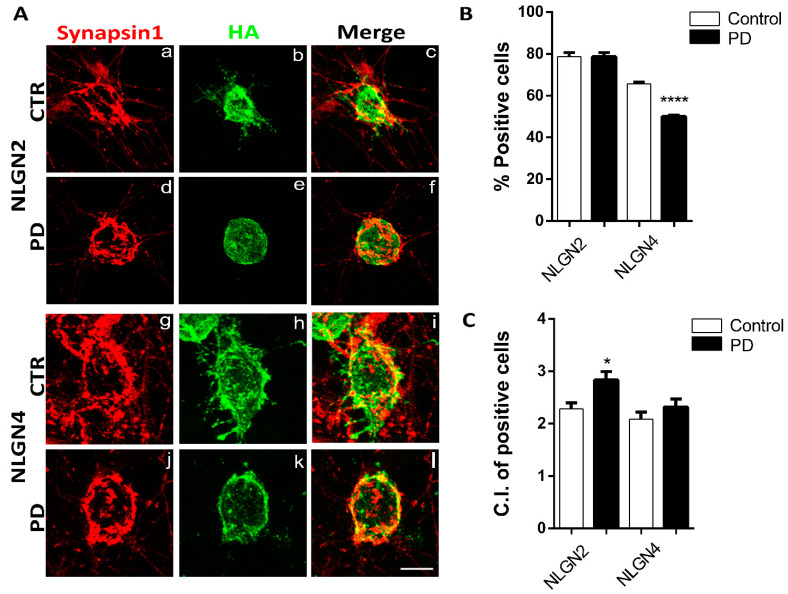
Heterologous synapse formation assays in PD neurons. (**A**) Representative immunofluorescence images of control and PD neurons co-cultured with HEK293T cells transfected either with NLGN2-HA (**a**–**f**) or NLGN4-HA plasmids (**g**–**l**). The co-cultures were stained with antibodies to HA epitope (green) and the presynaptic marker Synapsin 1 (red) and coincident green and red signals are shown in yellow. Scale bar = 50 µm (applies to all images). (**B**,**C**) Summary graphs showing the percentage of HEK293 positive cells for Synapsin 1 (**B**) and their C.I. of (**C**) when co-cultured with iPSC-derived neurons (NLGN2-HA Control: n = 68 cells of 85 cells/2 cultures; PD: n = 108 cells of 135 cells/3 cultures, NLGN4-HA Control: n = 50 cells of 77 cells/2 cultures; PD: n = 74 cells of 147 cells/3 cultures). All data represent mean ± SEM; statistical comparisons were made with two tailed *t*-test (* *p* = 0.01; **** *p* < 0.001).

## Data Availability

Gene Expression Omnibus (accession no. GSE84684).

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
