# Peer review of "Early Signs of Molecular Defects in iPSC-Derived Neural Stems Cells from Patients with Familial Parkinson’s Disease"

_biomolecules, 2022, doi:10.3390/biom12070876_

Round 1

Reviewer 1 Report

The article entitled ”Early signs of molecular defects in iPSC-derived neural stems  cells from patients with familial Parkinson’s disease” looks at differential gene’s expression in  iPSCs carrying A53T mutations.  The document is well written and overall, the result is presented clearly. A major point that should be discussed by the authors is the fact that the model they are using is inherently a neurodevelopmental model.  Perturbations of neurodevelopmental cellular functions and gene expressions would be expected. That doesn’t invalidate the findings, but this context is important with interpretation of the data.

Some specific minor points are made below;

-It will help with clarity if authors add more detail about the origins of the cells. Controls and A53T carrying.

For the introduction, term “Disease in a dish”, statement may give wrong impression of this model. These are not neurons in the CNS milleu. They are isolated developing neurons.

This needs  comment needs reference, “Third, compounds that are currently used for improving cognitive defects in AD and PD were highly effective in restoring neurotransmission in autism derived human neurons“

It’s not clear if they have anything to do with disease.

Figure 1 “folds expression” not clear what this unit is. Can authors use delta delta CT?

For results text. It is nice to include some statistical information. Telling us the FDR cutoff and such, so reader doesn’t have to search.

Author Response

We would like to thank the reviewers for the evaluation of our manuscript and constructive comments that we believe have addressed in our revised manuscript. All our changes are clearly marked as track changes in the manuscript. We also hereby provide a short description of the all changes and answers to both reviewers’ comments.

Reviewer 1

“It will help with clarity if authors add more detail about the origins of the cells. Controls and A53T carrying”. In Materials and Methods (section 2.1) a short description about the patient carrying the p.A53T mutation, the control subject, the origin of cell lines and the ethical approval was added according to published information included in Kouroupi et al, PNAS, 2017 (lines 82-90).

“For the introduction, term “Disease in a dish”, statement may give wrong impression of this model. These are not neurons in the CNS milleu. They are isolated developing neurons”. We have replaced “disease-in-a-dish” with “human iPSC-derived” that is more precise (Introduction line 49)

“This needs comment needs reference, “Third, compounds that are currently used for improving cognitive defects in AD and PD were highly effective in restoring neurotransmission in autism derived human neurons“”. The reference has been added (Introduction lines 60-62)

“It’s not clear if they have anything to do with disease”. We are unclear on where this comment refers to.

“Figure 1 “folds expression” not clear what this unit is. Can authors use delta delta CT?” In Results (section 3.1) Figure 1 was replaced with a new figure in which “Folds of Expression” was replaced with “Relative Expression”. Moreover, Figures 2 and 3 were replaced with new in which “Folds of Expression” was replaced with “Relative Expression”.

For results text. It is nice to include some statistical information. Telling us the FDR cutoff and such, so reader doesn’t have to search. In section 3.2 log fold changes (log FC) of the most important genes were added in brackets to be able to assess directly gene expression changes without the need of looking into the corresponding Supplementary Table.

A major point that should be discussed by the authors is the fact that the model they are using is inherently a neurodevelopmental model. Perturbations of neurodevelopmental cellular functions and gene expressions would be expected. That doesn’t invalidate the findings, but this context is important with interpretation of the data. This point is now addressed at the Discussion section (lines 396-399).

Reviewer 2 Report

The work is of interest, but there are some minor comments/questions:

 1.      The formatting of supplementary material (pages 4-46) should be improved, data are missing, it is difficult to follow these tables.

2.      There is no information about the patient (only in the supplementary file in reference 8). A short description would be advisable. The ethical permission is also not mentioned.

3.      For the sake of general readers, it would be worth mentioning the role of the investigated markers (paragraph 3.1, such as Nestin, Pax6, GFAP).

Author Response

We would like to thank the reviewers for the evaluation of our manuscript and constructive comments that we believe have addressed in our revised manuscript. All our changes are clearly marked as track changes in the manuscript. We also hereby provide a short description of the all changes and answers to both reviewers’ comments.

Reviewer 2

The formatting of supplementary material (pages 4-46) should be improved, data are missing, it is difficult to follow these tables. Supplementary Material, Tables 2 and 3 were replaced with tables that contain accurate information, are more clear and easy to follow and perform “search” by the reader.

There is no information about the patient (only in the supplementary file in reference 8). A short description would be advisable. The ethical permission is also not mentioned. In Materials and Methods (section 2.1) a short description about the patient carrying the p.A53T mutation, the origin of cell lines and the ethical approval was added according to published information included in Kouroupi et al, PNAS, 2017 (lines 82-90).

For the sake of general readers, it would be worth mentioning the role of the investigated markers (paragraph 3.1, such as Nestin, Pax6, GFAP). The roles of the selected markers are now described in the main text (lines 138-139, 144).